# Measurement of Melanocytic Choroidal Lesions: Ultrasound Versus Ultrawide-Field Fundus Imaging System

**DOI:** 10.3390/cancers17040642

**Published:** 2025-02-14

**Authors:** Maria C. Baradad-Jurjo, Daniel Lorenzo, Estel·la Rojas-Pineda, Laura Vigués-Jorba, Rahul Morwani, Lluís Arias, Pere Garcia-Bru, Estefania Cobos, Juan Francisco Santamaria, Carmen Antia Rodríguez-Fernández, Josep M. Caminal

**Affiliations:** 1Ophthalmology Department, Bellvitge University Hospital, Feixa Llarga Street, s/n, 08907 Hospitalet de Llobregat, Spain; dlorenzo@bellvitgehospital.cat (D.L.); rpstela@gmail.com (E.R.-P.); lauravigues@gmail.com (L.V.-J.); rahulmorwani89@gmail.com (R.M.); luisariasbarquet@gmail.com (L.A.); peregarciabru@gmail.com (P.G.-B.); ecobosmartin@gmail.com (E.C.); juanfra_s@hotmail.com (J.F.S.); carmenantia@gmail.com (C.A.R.-F.); jmcaminal@gmail.com (J.M.C.); 2Facultat de Medicina, Campus Bellvitge, Universitat de Barcelona (UB), Feixa Llarga Street, s/n, 08907 Hospitalet de Llobregat, Spain

**Keywords:** melanocytic choroidal tumors, basal diameter, ultrasonography, ultrawide-field imaging

## Abstract

Tumor size is an important factor in the diagnosis, treatment planning, follow-up and prognosis of melanocytic choroidal tumors. While ultrasonography is the most used technique for tumor measuring, new imaging techniques may also become a useful tool. The aim of our study was to assess the reliability of an ultrawide-field imaging system in measuring basal dimensions of choroidal tumors between observers and to compare this technique to conventional ophthalmic ultrasound. We found excellent concordance for interobserver and inter-technique agreement. Our findings suggest that, in some cases, ultrawide-field imaging could be used instead of ultrasonography for the measurement of basal diameters of melanocytic choroidal tumors. This technique is faster, more comfortable for the patient and less operator-dependent than conventional ultrasonography.

## 1. Introduction

Melanocytic choroidal lesions are the most common intraocular tumors [1,2,3,4,5]. Several clinical studies have found that tumor size is a key factor in the diagnosis of, selection of treatment for, follow-up of and prognosis of (i.e., larger melanomas are associated with higher mortality rates) these patients [6,7,8]. Currently, most ophthalmologists base their diagnosis on clinical examination, including indirect funduscopy and retinography, and ultrasound, which is the current gold-standard technique for measuring tumors, especially their thickness, as well as for tumor inner reflectivity characterization [9].

Several new imaging techniques have emerged in recent years, many of which have proven useful to better characterize these lesions and complement ultrasonography. For instance, MRI is becoming more popular and may give a more accurate volume calculation, but it is expensive and less immediate since it is not available in our daily ophthalmology practice [10,11,12,13]. With regards to ophthalmologic complementary testing, optic coherence tomography (OCT) and wide-field fundus imaging have an important role in our everyday practice [14,15,16]. The advent of noncontact ultrawide-field imaging systems has enabled the capture of the posterior pole and peripheral retina up to 200° in a single image. One example of this new technology is Optos^®^ (Optos, Marlborough, MA, USA), an ultrawide-field (UWF) scanning laser ophthalmoscope (SLO) that uses two laser wavelengths (red and green) to generate three different types of images: green laser view (green channel) images for the retina and its vasculature, red laser view (red channel) images for the choroid, and images with a combination of the two to digitally generate a pseudocolor realistic fundus image.

UWF SLO may prove to be a valuable tool for measuring basal tumor diameters, offering several advantages over ultrasonography such as being more comfortable for the patient and less time-consuming. However, it is necessary to consider that fundus imaging is a two-dimension representation of a three-dimensional structure obtained through the use of ellipsoidal mirrors. This results in some image distortion, especially in the peripheral retina, which can be magnified up to two times, and in the horizontal axis, which can appear stretched compared to the vertical axis [17]. Nevertheless, the new software version (ProView for Optos^®^ California, https://www.optos.com/our-software-products, access date 13 September 2024) includes a change in the algorithm to correct for the peripheral distortion that could make measurements more accurate [18].

The aim of our study was to assess the reliability of Optos^®^ California ultrawide-field (UWF) imaging in measuring basal dimensions of choroidal tumors between observers and to compare this technique to conventional standardized 20 MHz ophthalmic ultrasound (US).

## 2. Materials and Methods

### 2.1. Study Design and Eligibility Criteria

This was a retrospective study of data obtained from patients who underwent ophthalmological evaluation from January 2019 to January 2021 at the Ocular Oncology Unit of the Ophthalmology Department at Bellvitge University Hospital (Barcelona, Spain).

We included a consecutive series of patients diagnosed with untreated melanocytic choroidal lesions (either nevus or melanoma) that had undergone both UWF imaging and ultrasonography during the initial visit to our department. Patients that did not have the two imaging modalities performed on the same date and those without clear visualization of tumor borders in both imaging modalities were excluded. Patients excluded due to poor tumor visualization were classified in the following categories: peripheral tumors with partial visualization of the anterior margin, extensive exudative retinal detachment that blurred the visualization of the tumor, big tumors in which the apex covered the base partially, and tumors in which ultrasound was unable to provide an accurate assessment of the tumor size due to its low height.

### 2.2. Ultrasound Image Acquisition

All patients underwent a complete ophthalmologic ultrasound examination. In all cases, A- and B-scan images were obtained using a 20 Mhz US transducer (OTI-scan 2000, OTI Ophthalmic Technologies Inc.; North York, ON, Canada). Each examination included a longitudinal and transverse scan of the tumor. Calipers were used to measure the maximum transverse and longitudinal diameters on the B-scan image. All measurements were made by the same senior ophthalmology consultant (JMC).

### 2.3. Ultrawide Field Image Acquisition

For UWF imaging, using the Optos^®^ California (Optos, Marlborough, MA, USA) pseudocolor composite, red laser and green laser photos were obtained on the same day as the ultrasound by a technician. Transverse and longitudinal diameter measurements were assessed retrospectively for purposes of the present study by one ophthalmologist (observer 1) and one ophthalmology resident (observer 2), both blinded to the other’s measurements (both for UWF imaging and ultrasound). The basal diameters on UWF imaging were measured according to standard topographic US reference diameters (Figure 1).

### 2.4. Statistical Analysis

The demographic and clinical characteristics of the sample were described as means ± standard deviation (minimum–maximum) for quantitative variables and as numbers (n) and percentages for categorical variables. In all continuous variables used for comparison, the normal distribution was assessed by the Shapiro–Wilk test.

The following analyses were conducted to determine agreement between the devices and the observers with regards to lesion basal diameters (longitudinal and transverse diameters): (1) Paired *t*-tests and Bland–Altman plots were used to assess interobserver agreement for lesion size measured by UWF imaging and agreement between measurements of the lesion dimensions made by UWF (observer 1) and 20 MHz US (2). The interobserver reliability of UWF image measurements and the inter-technique reliability between the use of UWF and 20 MHz US were assessed using the intraclass correlation coefficient (ICC). The agreement was classified as poor (<0.6), good (0.6 to 0.74), or excellent (≥0.75).

All analyses were conducted using Stata IC, v. 15.1 (StataCorp; College Station, TX, USA). A two-tailed *p*-value < 0.05 was considered statistically significant.

## 3. Results

Images of 106 patients were reviewed. The initial diagnosis was distributed as follows: choroidal nevi (n = 44; 41.5%) and choroidal melanoma (n = 62; 58.5%). Of the 106 patients primarily included, 61 patients (57.5%) were excluded (20 choroidal nevi, 32.8%; 41 choroidal melanoma, 67.2%). The main cause for exclusion was, in 43 cases (70%), incomplete visualization of the tumoral base in UWF pseudocolor imaging due to the following reasons: peripheral tumors with partial visualization of the anterior margin (23 patients, 38%); extensive exudative retinal detachment that blurred the visualization of the tumor (9 patients, 14%); big tumors in which the apex covered the base partially (5 patients, 8%). In the remaining patients who were excluded from the study (18 patients, 30%), ultrasound was unable to provide an accurate assessment of the tumor size due to its low height. Eventually, 45 cases (42.5%) were included in the comparison of the two diagnostic techniques. Table 1 summarizes the baseline demographic and clinical characteristics of the patients, including ultrasound mean measurement values.

Firstly, interobserver agreement in UWF imaging was assessed. Table 2 shows the mean lesion sizes obtained by each observer and paired *t*-test results. Paired *t*-tests revealed that measurements from the pseudocolor and red channel images for both diameters were not statistically different when measured by the two observers (*p* > 0.05). However, for the green channel images, both observers stated that in 20 cases, the tumor could not be measured and, when it was possible, the values were statistically different (*p* < 0.05). For this reason, green channel images were not compared further. Figure 2 shows Bland–Altman plots for interobserver agreement in pseudocolor and red channel UWF imaging. The mean difference was around zero (between −0.28 and −0.13) in all cases, and 95% of the measurements fell within the limits of agreement except for the longitudinal pseudocolor images, where 6.7% of the points of the longitudinal basal diameter measurements were located outside the 95% limits of agreement.

Inter-technique agreement was then evaluated. Again, no statistical differences were detected between US and both pseudocolor and red laser photographs, while statistically significant differences were found on paired *t*-tests for the green channel images and US measurements (Table 3). Bland–Altman plots for the agreement in lesion size parameters between UWF pseudocolor and red laser images and US are shown in Figure 3. Even though both techniques were found to be significantly equivalent, ultrasonography mean values tended to be slightly higher than UWF values.

Finally, the correlation (ICC) between measurements was assessed. The UWF SLO (pseudocolor composite and red channel) between two observers (reliability of measures between observers) and the correlation (ICC) between measurements made with UWF SLO versus 20 MHz US (reliability of measures between tests) by specific measurements were excellent for all measures (Table 4).

## 4. Discussion

Ultrasound is currently the most frequently used tool to objectively measure tumor size, monitor growth and plan for the treatment of melanocytic choroidal tumors. However, while it is the gold-standard technique for tumor thickness measurement, clinicians may use and combine different imaging methods for basal diameter measurement [9,19]. Basal diameters are an important parameter for both diagnostic and prognostic reasons [6,7,8], and although US has shown to be a reliable technique when performed by an experienced ocular oncologist, it has some remarkable disadvantages [20,21]. For instance, US requires a certain tumor height (at least 0.4 mm) to allow the clinician to identify the tumor during examination, so quantification of the size of these tumors needs alternative imaging techniques. Moreover, even in larger tumors, the flat portion in the peripheral part of the lesions cannot be accurately assessed with US. On the other hand, ultrasonography examination is a time-consuming procedure that must be performed by a highly skilled professional and, nevertheless, repeating the same exact scanning plane in successive evaluations is virtually impossible. Finally, it is well known that ultrasound tends to overestimate dimensions when compared to histopathologic measurements, as reported in the COMS study, with particularly poor agreement for basal diameters [22].

Fundus photography has always been an important tool for the documentation of tumor characteristics and changes during follow-up. Over the past decade, advances in technology have made it possible to progressively increase the field of view of different imaging systems, so that lesions located in the peripheral retina, which could not be imaged with traditional fundus photography, can now be visualized with these new devices [23]. Among other advantages, this type of examination is comfortable for the patient, as it often does not require mydriasis, is performed by a trained technician, and may be suitable for telemedicine. Regarding the assessment of basal tumor dimensions using UWF funduscopic images, the main advantage over ultrasound is the reproducibility of the examination and the ease of locating lesions and repeating measurements in the same plane in successive examinations as well as measuring the distance to important intraocular structures such as the fovea or the optic nerve head.

Nonetheless, the main drawback of assessing tumor basal dimensions using UWF funduscopic images arises from the fact that it is a representation of the curved surface of the retina displayed on a flat plane. Distortion of the peripheral retina results in variable magnification of the lesion size when it is located post-equatorially depending on eye characteristics, such as axial length, refraction or different lesion location, and the device used [24,25]. In order to minimize peripheral distortion, improvements in image processing software have been attempted. For the purpose of our study, we used the Optos^®^ California device, which includes ProView (https://www.optos.com/our-software-products, access date 13 September 2024), upgraded software, which corrects the peripheral distortion and horizontal elongation found in its predecessor, the Optos^®^ 200TX [18,26].

The aim of our study was to assess the reliability of the Optos^®^ California UWF imaging system in measuring dimensions of choroidal melanocytic tumors between observers and to compare and correlate this technique with 20 MHz US.

We found an excellent ICC and defined Bland–Altman plots for interobserver and inter-technique agreement in estimating basal diameters when using pseudocolor composite and red laser images. However, when using green laser images, agreement was poor and, in many cases, measurement was not possible. As stated before, the green laser cannot penetrate below the retinal pigmented epithelium (RPE) and, thus, the image offers a good view of the retina and its vasculature but does not allow for visualization of the choroid. Although retinal changes are observed in many melanocytic choroidal tumors and they translate into alterations in the green channel images, these differ in each tumor and are not related to its size. On the other hand, the red laser has the ability to penetrate through the RPE and gives a clearer image of the choroidal tumor boundaries. Overall, this is consistent with the fact that agreement is good when using the red laser images or its pseudocolor composite, but it becomes unreliable when using the green channel. Although it did not give statistically different results in our study, the red channel should theoretically provide more accurate measurements than the pseudocolor composite since the latter could be influenced by retinal changes detected by the green laser.

A few studies have previously compared US and UWF imaging systems in the measurement of choroidal tumors [27,28,29]. Our findings align with those of previous studies, yet comparisons are challenging due to the use of different UWF imaging devices and the inclusion of participants with diverse characteristics, including those with choroidal tumors of varying histological origins. The study by Ayres et al. [29], which included melanocytic and non-melanocytic choroidal tumors measured by US and Optos^®^ 200TX, concluded that US and UWF pseudocolor and red laser images were statistically equivalent. In another study, Wang et al. [30] used Clarus 500 with similar results. Although no statistical differences were found, both studies reported a tendency toward higher values for UWF imaging in comparison to US. In our study, however, we did not encounter such a tendency, and, in general, the US measurements were slightly higher than the UWF values. This is likely attributable to the aforementioned software upgrades that address peripheral magnification.

The main limitation in the use of UWF imaging in the measurement of the basal diameters of melanocytic choroidal lesions is the large number of cases with incomplete recognizable borders (57% in our study), mainly due to the partially visible anterior margin in peripheral tumors. In these cases, as well as in cases with any other cause of improper visualization of the tumor margins such as media opacity or exudative retinal detachment, ultrasonography continues to be essential. In contrast, in those tumors with a low height that cannot be located by US, UWF imaging has become a useful tool for both documentation and sizing.

The main limitation of this study is the relatively limited number of tumors evaluated, the large number of excluded cases, and the inherent limitations of it being a retrospective study. The main strength of the study is that this is the first to compare Optos^®^ California and its improved software ProView to 20 MHz US to measure the basal diameters of melanocytic choroidal tumors, as well as the fact that it had a more homogeneous sample than previous studies that have included choroidal tumors of varying histological origins.

## 5. Conclusions

While ultrasonography is essential in the measurement of melanocytic choroidal tumors, our findings suggest that when complete visualization of the tumor is possible, UWF fundus imaging could be as reliable as ultrasonography in the measurement of the basal diameters. An important advantage of UWF imaging versus US is that it is less operator-dependent and more comfortable for the patient, which makes it a desirable characteristic for routine clinical use. However, while both techniques can be complimentary, ultrasonography cannot be replaced by UWF fundus imaging since it remains the gold-standard technique for tumor thickness measurement and it is the only option in cases where there is unclear visualization of the tumor margins or media opacity.

## Figures and Tables

**Figure 1 cancers-17-00642-f001:**
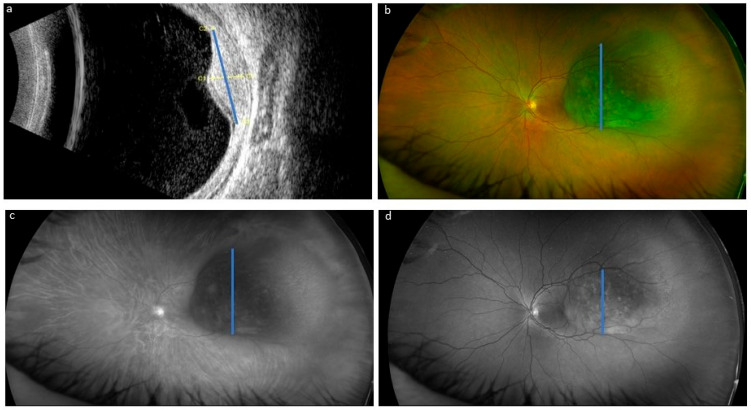
Example of measurement of choroidal melanoma transversal diameter (blue line) by ultrasonography (**a**), ultrawide-field pseudocolor composite (**b**), red laser imaging (**c**) and green laser imaging (**d**). In this case, we found a difficulty in locating the border of the tumor in the images of the green channel with respect to those of the red channel and those taken using the pseudocolor composite.

**Figure 2 cancers-17-00642-f002:**
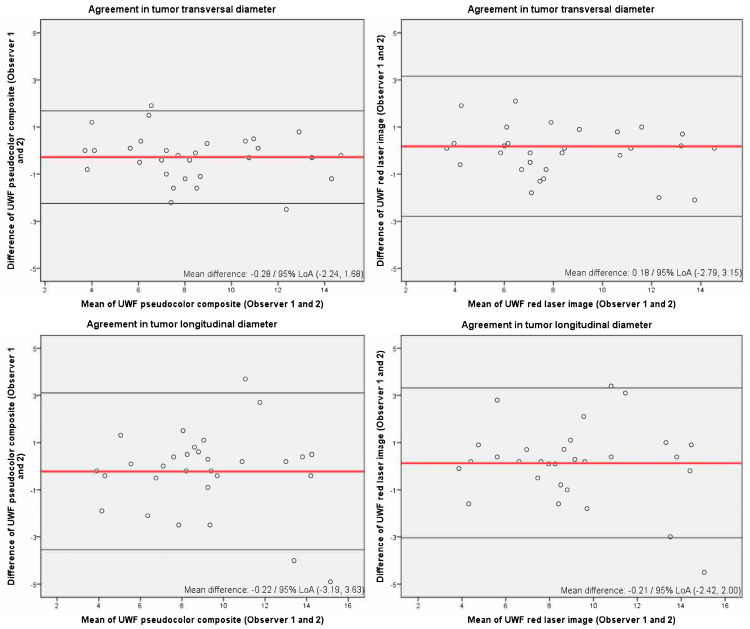
Bland—Altman plots for inter-observer agreement in tumor size parameters using utra-wide-field imaging. To aid interpretation, the same scale is used in all plots. LoA: limits of agreement.

**Figure 3 cancers-17-00642-f003:**
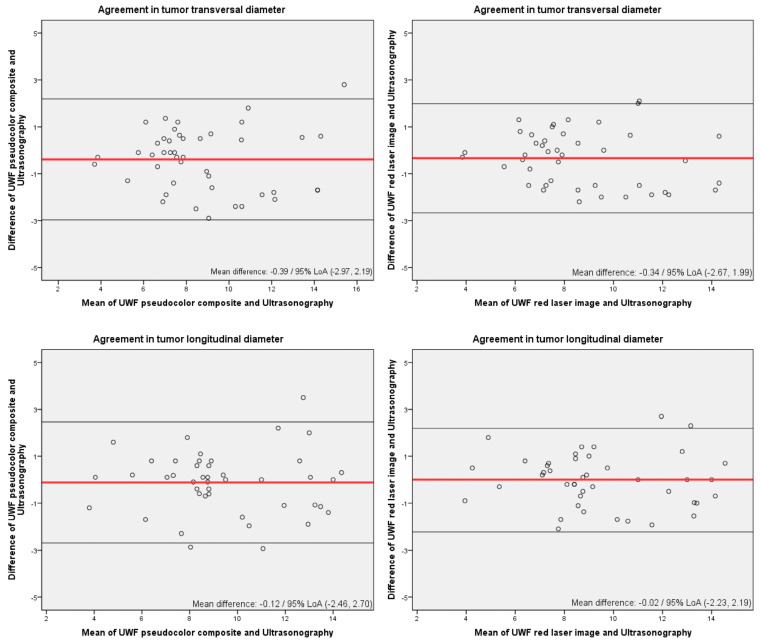
Bland–Altman plots for the agreement in lesion size parameters between ultrawide-field imaging and 20 MHz ultrasonography. LoA: limits of agreement.

**Table 1 cancers-17-00642-t001:** Baseline features of included patients.

Demographic characteristics	
Mean Age (Years)	67 ± 12.6 (38 to 87)
Gender (Female/Male)	27 (60%)/18 (40%)
Eye (Right/Left)	25 (56%)/20 (44%)
Clinical characteristics	
Best corrected visual acuity—decimal scale	0.5 (Count Fingers to 1.0)
Diagnosis	
Nevus	24 (53%)
Melanoma	21 (47%)
Pigmentation	
Pigmented	44 (98%)
Amelanotic	1 (2%)
Subretinal fluid	
Present	24 (53%)
Absent	21 (47%)
Tumor location	
Posterior pole	11 (24%)
Periphery	34 (76%)
Ultrasound measurements	
Height (mm)	2.65 ± 1.51 (0.70 to 6.10)
Transversal diameter (mm)	8.84 ± 2.91 (4 to 15)
Longitudinal diameter	9.36 ± 2.90 (4 to 14.50)

Values represent mean ± standard deviation (minimum and maximum) value for quantitative variables, and n (percentage) for categorical variables.

**Table 2 cancers-17-00642-t002:** Mean values in mm and standard deviation (SD) for each diameter measured by the two observers in UWF fundus images, and paired *t*-test comparison results of the different measurements.

	Observer 1	Observer 2	*p*-Value (Paired *t*-Test)
Transversal diameter—pseudocolor composite	8.45 (2.86)	8.64 (2.98)	0.126 *
Longitudinal diameter—pseudocolor composite	9.24 (2.90)	9.39 (3.01)	0.471 *
Transversal diameter—red channel	8.38 (2.60)	8.25 (2.87)	0.491 *
Longitudinal diameter—red channel	9.30 (2.86)	9.21 (3.07)	0.641 *
Transversal diameter—green channel	7.11 (3.99)	7.48 (3.87)	0.038
Longitudinal diameter—green channel	8.15 (4.13)	8.42 (3.98)	0.044

Non-statistically significant difference between the means (*).

**Table 3 cancers-17-00642-t003:** Paired *t*-test comparison results of the two measurement methods.

Measurement Technique	Ultrasound
UWF TD—pseudocolor composite	0.051 *
UWF LD—pseudocolor composite	0.525 *
UWF TD—red channel	0.063 *
UWF LD—red channel	0.870 *
UWF TD—green channel	0.000
UWF LD—green channel	0.000

UWF: ultrawide field. TD: transversal diameter. LD: longitudinal diameter. * Non-statistically significant difference between the means.

**Table 4 cancers-17-00642-t004:** Intraclass correlation coefficient (95% confidence interval) between measurements made with UWF between two observers (reliability of measures between observers) and results of the ICC between measurements made with UWF versus 20 MHz US (reliability of measures between tests) by specific measurements.

**ICC between measurements made with UWF between two observers**		
	Transverse diameter	Longitudinal diameter
Pseudocolor composite	0.98 (0.96–0.98)	0.94 (0.89–0.97)
Red channel	0.95 (0.90–0.97)	0.95 (0.90–0.97)
**ICC between measurements made with UWF versus 20 MHz US**		
	Transverse diameter	Longitudinal diameter
Pseudocolor composite—US	0.94 (0.89–0.97)	0.95 (0.90–0.97)
Red channel—US	0.95 (0.90–0.97)	0.96 (0.93–0.98)

## Data Availability

The raw data supporting the conclusions of this article can be made available by the authors on request.

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
