# Peer review of "Measurement of Melanocytic Choroidal Lesions: Ultrasound Versus Ultrawide-Field Fundus Imaging System"

_cancers, 2025, doi:10.3390/cancers17040642_

Round 1

Reviewer 1 Report (Previous Reviewer 3)

Comments and Suggestions for Authors

The last corrections are useful and clarify that Ultrasonography cannot be substitute by ultrawide field (UWF) imaging, because the main parameter considered for detecting uveal neoformations is their thickness and thickness can't be evaluated with UWF imaging                                      

Author Response

Dear reviewer,

Thank you for your feedback. We are glad to hear that the previous corrections address your concerns. We have made additional changes to the conclusion in order to clarify the role of both techniques as you have stated in your last review. 

Thank you for your time and consideration.to clarify the role of both techniques, as you

Sincerely,

Maria C Baradad Jurjo

Reviewer 2 Report (Previous Reviewer 2)

Comments and Suggestions for Authors

thanks, I think, it is OK now.

Author Response

Dear reviewer,

Thank you for your feedback. We are glad to hear that the previous corrections address your concerns stated in the previous review. 

Thank you for your time and consideration. 

Sincerely, 

Maria C Baradad Jurjo

This manuscript is a resubmission of an earlier submission. The following is a list of the peer review reports and author responses from that submission.

Round 1

Reviewer 1 Report

Comments and Suggestions for Authors

I agree with the authors ultrasound, indirect ophthalmoscopy and fundus images are crucial to measure intraocular tumors, even in expert eyes.

"While ultrasonography is the gold standard technique for tumor measuring..." needs to be improved. Ultrasonography is the gold standard to measure tumor's thickness, not diameter since there is a flat portion of tumors that ultrasound is not useful. This concept is present in the entire manuscript and should be corrected.

Figure 1 (a and b): it can be argued how both ultrsound and Optos image measure the tumor in these figures.

Line 178: "Ultrasound is currently the gold standard tool to measure tumor size, monitor growth and plan treatment of melanocytic choroidal tumors" needs to be changed: the combination of indirect ophthalmoscopy, fundus photography and ultrasound is the gold standard of measuring intraocular tumors.

Major revision is needed.

Author Response

Dear reviewer,

Thank you for your feedback, we appreciate the time you have taken to provide these valuable comments. After reviewing them, we have made some changes and would like to discuss some points.

Firstly, we completely agree, and indeed it is the purpose of our manuscript, that while ultrasonography is the gold standard technique for thickness measurement, when it comes to basal diameters combination with other techniques such as fundus photography is essential. This is obvious for qualitative evaluation of the lesions and, with our study, we would like to point out that quantitative measurement regarding basal diameters is also possible and similar to ultrasonography. While there is no well stablished gold standard for basal diameter measurement, it is true that ultrasonography is widely used and many ophthalmologists stablish the tumor diameter to plan treatment and follow up with ultrasound measurements alone. In this regard, and in order to make our message clearer, changes have been made in the simple summary as well as in lines 42-45, 186-189, 195-196 and 276-278.

Regarding comparison of ultrasound and Optos measurements and how the measurements are performed, we have assessed the interobserver reliability when measuring the tumors with Optos and the result has been excellent .In regards to ultrasonography, the measures were taken by a clinician in our oncology department with wide expertise in ultrasonography examination and following the standardized transversal and longitudinal planes.

Thank you for your time and consideration.

Sincerely,

Maria C Baradad Jurjo

Reviewer 2 Report

Comments and Suggestions for Authors

Please, add more comparisons to other methods e.g. volume measurement of the tumor by MRI, or CT.

Please, give exclusionary criteria for UWF imaging measurements.

Author Response

Dear reviewer,

Thank you for your feedback, we appreciate the time you have taken to provide these valuable comments and the opportunity to submit a reviewed draft of our manuscript.
Regarding point 1, while we could talk extensively about tumor volume calculation with MRI and how it compares to ultrasound volume calculation, we have found no papers connecting this technique with UWF funduscopy so we find it difficult to link extensive information about MRI with our paper. However, we completely agree that MRI is useful and an increasing number of ocular oncologists are using it for volume calculation so we have found very adequate to include a comment and references in the second paragraph of the introduction.

Regarding point 2, the exclusionary criteria were only mentioned in the results section and now we have included specific exclusionary criteria in the methods section (lines 81-86).

Thank you for your time and consideration.

Sincerely,

Maria C Baradad Jurjo

Reviewer 3 Report

Comments and Suggestions for Authors

On my opinion ultrasounds always remain the gold standard  for melanocytic tumors, because are the only tool that permit a correct evaluation of the tumor thickness and of the tissue characteristics  and not only of the basal diemeters and it is much more useful in evaluating peripherical lesions (57% of cases excluded due to the peripheral position). Only in some cases (flat and non evolutive lesions), it is possibile to replace a conventional ultrasound exam with ultrawidefield imaging. The Authors have to explain very well this message in the paper, otherwise the readers could have a partial information.  

Author Response

(The authors gave the same response as above.)

Round 2

Reviewer 3 Report

Comments and Suggestions for Authors

On my opinion it is not possible to make a correct diagnosis of uveal melanoma without ultrasonography, even if is a time consuming technique and just with the basal diameters of the lesion, because the tumor thickness and its inner status  evaluation (standardized A-Scan) are essential for it. So the ultrawidefield examination is an important and useful tool for the assessment of this kind of lesions, but it can never take the place of ultrasonography in every suspect of malignancies, but just in case of clear nevi or other benign and flat tumors with completely evaluable edges, less than 50% of the paper sample.